# Water Savings, Yield, and Economic Benefits of Using SRI Methods with Deficit Irrigation in Water-Scarce Southern Iraq

**Mohammed Khalid Mohammed [1,\*], Khidhir Abbas Hameed [2] and Abdulkadhim Jawad Musa [2]**

[1]   Department of Economic Research, Ministry of Agriculture, Baghdad 10011, Iraq
[2]   Al-Mishkhab Rice Research Station, Ministry of Agriculture, Najaf 54001, Iraq
\*   Correspondence: moh_mmed85@yahoo.com; Tel.: +964-790-674-1411

**Abstract:** This study evaluated what intervals of irrigation in conjunction with the use of SRI methods could achieve the greatest economic, as well as agronomic returns when growing irrigated rice under the water-deficit conditions of southern Iraq. A field study at the Al-Mishkhab Rice Research Station in southern Iraq recorded input and output data for three different irrigation regimes: continuous submergence of the rice crop; irrigation at 3-day intervals; and irrigation at 7-day intervals. Benefit–cost analysis showed 3-day intervals with SRI methods, giving the highest net returns and highest internal rate of return, indicating that the continuous irrigation of rice fields is a waste of water, with neither agronomic nor economic benefit. In Iraq, there are large opportunity costs for any unnecessary use of irrigation water. The highest water productivity was achieved with 7-day intervals of irrigation together with SRI methods, but this entails some sacrifice of the yield ha$^{-1}$, as 13% less grain is produced than with continuous submergence of the crop. With 7-day intervals compared to 3-day intervals, water-saving was 44%, but compared to continuous submergence of the crop, the saving was 72%. This large amount of water could, if redeployed, enable many more farmers to cultivate larger areas of the land, increasing total rice production for Iraq, and some of the water saved could be put to other, high-value uses. It would thus benefit the country and many farmers if, in return for rice farmers' using irrigation water more productively, those who currently grow rice could be persuaded to accept a grain yield somewhat lower than they could produce with 3-day intervals of irrigation and SRI crop management. The redeployment of water saved by having longer irrigation intervals coupled with SRI methods could raise Iraq's rice output more than enough to compensate the farmers for forgoing some attainable production by their accepting 7-day irrigation intervals. If no such incentive scheme could be established, there would still be a significant benefit for farmers and for the country by moving to SRI production methods with 3-day intervals rather than continuing the present practice of routinely flooding rice fields and using conventional methods.

**Keywords:** water saving; system of rice intensification; water-deficit irrigation

## 1. Introduction

Rice is the most important main-season crop in Iraq, occupying about 5–6% of its cultivated area and being a strategic crop necessary for food security. All land in Iraq is precious because only about 11–12% of its total land area is arable. In 2018, Iraqi farmers were producing almost 500,000 tonnes of paddy rice from 128,000 ha, thus with an average yield of about 4.5 tonnes ha$^{-1}$. Najaf and Diwaniya are the most important governorates for growing paddy rice, as their planted area constitutes almost 70% of the total area planted in paddy in Iraq. These two governorates account for about 80% of the country's total paddy rice production because the farmers there have the best access to irrigation services [1].

In Iraq, water is a greater constraint on agricultural production than land [2]. Current water use in rice production is 3073 m$^3$ of water per kg of rice produced, more than twice the global average of 1325 m$^3$ per kg [3]. This reflects the high rate of evaporation due to

elevated temperatures during most of the year, and the high rates of seepage that result from Iraq's porous soils, which, for the most part, have little organic matter [4].

Because rice is such a strategic crop, it is subsidized by the state, which guarantees to farmers that their crop will be purchased at a reasonably set price. The international market price for rice does not affect the price in local markets in Iraq because the government monopolizes paddy purchases and maintains a public food ration [5]. Recently, although a minimum price is still guaranteed, producers are free to market their product either to the state or the private sector if its market prices are higher than those offered by the state.

Most Iraqi rice farmers cultivate rice according to the cultural practices inherited from their parents before them. With traditional rice cultivation, rice crops are established with direct sowing. This uses large amounts of seed, about 200 kg ha$^{-1}$, which is many times more than with SRI [6]. Additionally, throughout the crop cycle, standing water is maintained on the soil surface of rice paddies to a depth of 5–20 cm [7]. With such practices, a lot of water is consumed without producing very much yield. Accordingly, there is a need in Iraq for a new strategy to achieve higher productivity of rice with less water.

With SRI practices, plant roots grow larger and deeper and do not degenerate due to a lack of oxygen in the soil, which occurs when rice fields are kept continuously flooded. Moreover, with no continuous inundation there is more beneficial life in the soil, which requires oxygen for its maintenance. SRI practices working together elicit more profuse and more vigorous growth of plant roots.

When rice plants are grown under submerged conditions, their roots are deprived of oxygen and degrade over time, while when they are grown with alternate wetting and drying (AWD), there is little or no root deterioration [8]. When rice plants experience intermittent water stress, this signals them to grow more and deeper [9]. On the other hand, when plants are continuously flooded and supplied with inorganic fertilizer, they do not need to extend their root systems downward because they have both a continuous supply of water and superficially accessible nutrients. Inundated rice plants have little need to grow deeper, more profuse roots, and also less ability to do this because of the anaerobic soil conditions. The roots of SRI plants with alternate wetting and drying (AWD), on the other hand, can better access nutrient stocks and water in the lower horizons of soil by growing more deeply.

Globally, irrigated areas constitute only about 17% of the land area devoted to producing food; however, they provide >40% of the global food supply [10]. Currently, the supply of water for irrigated agriculture is limited, but it will be even more limiting in the future. Irrigated agriculture in many areas in the world is practiced without concern for the basic principles of resource conservation and sustainability. Irrigation in areas of water scarcity needs to be carried out with utmost efficiency, both saving water and maximizing its productivity, getting more crop per drop. With SRI, farmers increased grain yield with less water, less seed, and less need to purchase [11].

This paper examines how practicing SRI crop and water management methods with different intervals of irrigation could raise both the crop yield and water use efficiency, as both are needed under Iraqi conditions [12]. Despite the common belief being the contrary, rice is not an aquatic plant that performs best under continuous flooding, because its root systems degrade under hypoxic conditions; but how much water limitation or stress can rice plants tolerate without sacrificing yield? We wanted to evaluate this in a water-stressed environment, also looking at the economics of reducing water consumption for farmers and for the country.

## 2. Materials and Methods

### 2.1. Material Collection

This paper is based on the input and output data collected from on-station trials where SRI methods of rice production were used with three different irrigation regimes at the Al-Mishkhab Rice Research Station in the Najaf province of southern Iraq. The climate there is hot and dry, with high rates of evaporation during the summer growing season

(7.38 mm d$^{-1}$). Summertime temperatures range between 28.2 and 45.6 °C [13]. The texture of the soils is silty clay loam to clay loam (silt 48.7%; clay 31.3%; sand 20.0%), with a pH of 8.0 and an EC of 2.1 ds/m according to field soil analysis.

### 2.2. Study Design and Operation

The irrigation regimes evaluated were: (a) continuous submergence of the rice crop throughout the growing season using current cultural practices, as described below; (b) issuing irrigation water at 3-day intervals, with SRI practices; and (c) issuing water at 7-day intervals, with SRI practices. The fields were leveled and managed as plots, each 1000 m$^2$ in size (10 m × 100 m). The whole plot was harvested and weighed separately, not sampled, to calculate the grain yield with each of the irrigation treatments adjusted to hectares.

The rice variety used on all of the plots was a local variety of high-yielding, aromatic Jasmine rice, with medium growth duration and resistance to lodging. The SRI fields were planted with young seedlings 15 days old, transplanted by hand, one seedling per hill, with a 25 cm distance within and between the rows. An example from our research station is seen below in Figure 1. These two rice plants are the same age and variety, but they are grown with different methods: SRI on the left, and farmers' current practices on the right.

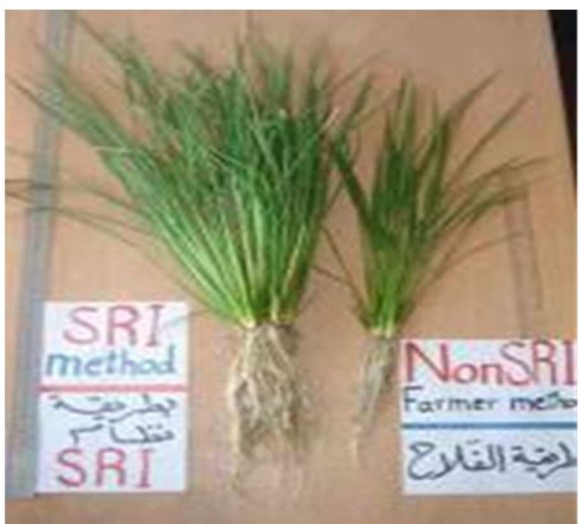

**Figure 1.** Rice plants grown at the Al-Mishkhab Rice Research Station, Najaf, Iraq (Jasmine variety), in the 2010 main season (@ 60 days).

Water was supplied through three intermittent irrigations during the vegetative phase, and then continuous flooding with just a thin layer of water (1–2 cm) after panicle initiation phase.

For soil fertilization, in addition to providing 10 tonnes of compost ha$^{-1}$, half of the usual recommended amount of inorganic fertilizer was applied, i.e., 200 kg ha$^{-1}$ of granular NP mixed with soil, and 160 kg of urea ha$^{-1}$. Weeds were controlled twice by hand. Unfortunately, it was not possible to do soil-aerating mechanical weeding as recommended for SRI, as no appropriate weeders were available. This practice could have been expected to increase the yield of the SRI plots.

The conventional management practices with which the SRI was compared are sometimes referred to as dry-method practices. Seeds are broadcasted directly onto ploughed land using a large amount of seed (as much as 200 kg ha$^{-1}$). Chemical fertilizer was applied (400 kg ha$^{-1}$ of compound NP [18 × 18] and 280 kg of urea ha$^{-1}$), along with 10 L ha$^{-1}$ of Stam F34, a post-emergence herbicide, for weed control. In addition, hand-weeding was carried out 3–4 times, as weeds began to grow and appear on the surface of the soil at different stages due to the different depths of their site in the soil. The use of the pesticide just one time was not enough. Organic manure was not provided, and the soil was kept

continuously submerged with a layer of water covering the surface until the crop reached maturity. At maturity, the whole plots were harvested to evaluate grain yield.

For the first 10 days after transplanting, water was applied to all plots in a similar manner, i.e., one irrigation issue of 7.5 cm depth was given daily to ensure good seedling establishment. With the continuous submergence method, water was applied daily thereafter, maintaining a layer of water 10 cm deep in the plot during the entire growing season. For the intermittent irrigation treatments, there were intervals of either 3 or 7 days between the successive irrigations, with approximately 8.5 cm of water given at each irrigation. The amount of water used for the three methods of irrigation management is shown in Table 1, as measured by a standard water meter.

**Table 1.** Amount of irrigation water used with different irrigation regimes and SRI practices.

| Irrigation Method | Irrigation Water Used ($m^3$ $ha^{-1}$) | Water Use as % of CS | Water Saving (in %) |
|---|---|---|---|
| Continuous submergence (CS) | 79,090 | – | – |
| 3 d intervals | 39,485 | 50% | 50% |
| 7 d intervals | 22,072 | 28% | 72% |

*2.3. Data Analysis*

Data were collected on the amounts of inputs and outputs with each of the three methods, together with their respective market prices. From this we calculated the net income and benefit–cost ratio associated with each irrigation method together with either conventional or SRI practices. Benefit–cost analysis is the summary method for assessing net gains or net losses from an activity in economic terms. This ratio is relevant for both farmers and the country at large.

Beyond economic returns, it is important to consider what is the productivity of water, the scarcest and most limiting factor of production in Iraq. In a water-scarce environment, water use efficiency can be of equal or even greater concern than economic returns because there are significant opportunity costs for any inefficient use of water. Putting scarce water to its most productive uses has positive implications for the whole country.

**3. Results and Discussion**

*3.1. Agronomic and Water Productivity*

The findings of this study are summarized in Table 2. The grain yield of continuous submergence (CS) with SRI methods was 5.8 tonnes $ha^{-1}$. Readers should note that this is about 30% more than the current national average yield. This means that the comparisons with water-deficit irrigation made here are with a very respectable yield for Iraq, not with the low current yield attributable to poor management practices.

SRI-managed rice plots that received an issue of water only every 3 days had paddy yields 20% higher than the continuously flooded SRI plots, with just half as much water being provided. Accordingly, water productivity was more than doubled, i.e., increased by 240%. When irrigation water was issued to SRI-managed plots only every 7 days, on the other hand, the yield from these more water-stressed plants was 13% lower than with continuous submergence. However, this crop consumed only one quarter of the water required for continuous submergence.

This increase of 320% represents a tripling of water productivity compared to continuous submergence. Three quarters of the water that is presently used to keep paddy fields flooded throughout the growing season could be released from irrigated rice production with a relatively small reduction in the paddy yield per hectare, and a large increase in the rice that is produced per $m^3$ of water issued.

**Table 2.** Average grain yield and water productivity with SRI crop management and different irrigation methods.

| Irrigation Method | Grain Yield (t ha$^{-1}$) | % of CS | Water Consumption (m$^3$ ha$^{-1}$) | % of CS | Water Productivity (kg m$^{-3}$) | % of CS |
|---|---|---|---|---|---|---|
| Continuous submergence (CS) | 5.83 | – | 79,090 | – | 73.7 | – |
| 3 d intervals | 7.02 | +20% | 39,485 | -%50 | 177.8 | +240% |
| 7 d intervals | 5.20 | −11% | 22,072 | %72- | 235.7 | +320% |

The water saved could, at least in principle, be made available for other needs, expanding the area of land under irrigation and/or using the water saved for commercial, domestic, or other purposes. We say 'in principle' because water is a bulky commodity and cannot be moved without considerable cost, except by gravity. Thus, topography, logistics, and economics all need to be taken into account. However, a large amount of water could become available for other uses.

*3.2. Economic Evaluation*

The costs and returns associated with each of the three irrigation regimes evaluated are laid out in Table 3. This comparison is similar to the productivity made in Figure 1, reflecting the differences that farmers would experience in their respective incomes per hectare from the three different methods of water management for trice production being evaluated. There are obvious advantages from using SRI methods with 3-day intervals of irrigation in terms of grain production, water-saving, etc.

**Table 3.** Comparison of the costs and returns from growing rice with alternative irrigation regimes when using SRI methods for crop management (in dollars/ha$^{-1}$).

| Irrigation Method | Costs of Production | Value of Production (Rice Market Value) | Net Economic Returns (Income–Costs) | Change from CS |
|---|---|---|---|---|
| Continuous submergence (CS) | 1208 | 3158 | 1950 | —— |
| 3 d intervals | 1166 | 3803 | 2637 | +35% |
| 7 d intervals | 1116 | 2818 | 1702 | −13% |

Notes: Production inputs include seed, fertilizer, pesticides, electricity, fuel, transport, machinery, field preparation, and repairs. The costs of harvesting are not included. They would be somewhat higher for 3-day intervals and lower for 7-day intervals because of differences in yield.

However, this analysis does not consider the full costs or all benefits, such as opportunity costs (e.g., economic and other effects beyond the farm level) for the government and nation, other farmers, entrepreneurs, and the citizenry at large. A big omission in this analysis is that it assesses the benefits and costs from the perspective of farmers, the water users, rather than from society as a whole. The costs and value of water as a production input for rice farmers are not included in these calculations because farmers do not pay for their water in Iraq.

Even if irrigation water has no direct cost for farmers, it has much value as well as certain costs for society that should be taken into account for any full evaluation. For one thing, the government bears some costs for delivering water (free) to farmers. In Iraqi society, water is considered as coming from God or Nature, so people think that it should be free for everyone. The government expenditure for providing irrigation water gets justified by citizens' expectation that, when they pay their taxes and give their allegiance to the government, they will be able to have enough food at a reasonable price to sustain

themselves. There is thus an implicit social contract, with farmers as a sector and also with consumers, everybody in the country.

More broadly, we need to appreciate that in Iraq, all water, being scarce, has opportunity costs. There could be considerable benefits to society as a whole if scarce water can be put to higher-value alternative uses. Thus, for an inclusive economic analysis, the costs and/or cost-savings with regard to water should be worked into the calculations we have made. Such an analysis would require making many assumptions and valuations that are beyond our scope here. We are reporting data and analysis based on experimental evidence, hoping that this should encourage and provide a basis for broader policy considerations.

It is not our task to make recommendations for how gains from water-saving by using SRI methods with reduced water consumption should be distributed. However, we can lay out some alternative scenarios. The data clearly indicate that using SRI methods with any of the three irrigation regimes evaluated is preferable to the status quo. We take continuous irrigation with SRI methods as a baseline for comparison, keeping the analysis somewhat simpler than if all possible options are considered. The comparisons would be even starker if they were made with current rice-growing practice.

### 3.2.1. Shifting to SRI Methods with 3-Day Intervals of Irrigation

How much water saving would there be with this option? This would not reduce the government's direct costs and might even raise them a little to deliver less water in smaller, but reliable amounts to farmers, delivering irrigation water in a timely way. This would require some investment in improving the physical infrastructure for water control (hardware) and some expenditure for irrigation personnel and water user associations (software). However, given the large amount of water that could be freed up from rice production for other uses, this would be a justifiable expenditure (investment) from a societal perspective. There should be a considerable financial surplus generated by these changes that the government could use to meet other pressing needs.

The amount of expenditure required to improve irrigation control and distribution could be estimated in advance to assess, at least roughly, how much revenue and value could be generated beyond the farm level. It should also be considered what an increase in rice production would do to enhance national food security and increase farmers' net income for improving the welfare of rural communities.

There would be some cost for training farmers in SRI methods, but that should be a one-time investment, since the subsequent supervision of SRI should not entail additional costs because existing extension staff would be overseeing and supporting SRI instead of conventional rice cultivation, as they do now. The costs for making a change to SRI methods for rice production should be considerably less than the farmer and national benefits created.

### 3.2.2. Shifting to SRI Methods with 7-Day Intervals of Irrigation

The considerations for this alternative are more complicated because there would be less total rice produced from the present rice area, about 13% less, but a huge saving of water, three-fourths of the amount that is presently utilized for rice production. Some of this water could be used to expand the irrigation command area (service area) to grow more rice or other crops. This would increase food production and give more people access to irrigation and irrigated land for productive work and earning incomes.

The water saved could, in principle, permit as much as a quadrupling of the irrigated area. However, using additional water for irrigated agriculture would depend on topography and land suitability, and probably only some portion of the water saved should be utilized for increasing food production. The rest of the water saved could, at least in principle, be used for other purposes.

*3.3. Water Requirements for Different Extents of Rice Areas in Iraq*

Using data for 2019 from the Ministry of Planning's Central Statistical Organization, Table 4 shows calculations of the water requirement for each of the three irrigation methods to serve the full rice area cultivated in Iraq at present.

**Table 4.** Total water requirements for rice area cultivated in Iraq.

| Irrigation Method | Water Consumption (m³ ha⁻¹) | Rice Area Cultivated | Total Water Requirements (Billion m³) | Estimated Rice Output (Million Tonnes) |
|---|---|---|---|---|
| Continuous submergence | 79,090 | 127,800 | 40.4 | 741,240 |
| 3 d intervals | 39,485 | 127,800 | 20.2 | 894,600 |
| 7 d intervals | 22,072 | 127,800 | 11.3 | 664,560 |

By expanding the currently irrigated rice area by 14,750 ha (to 142,500 ha), a 7 d interval irrigation regime could match the production from SRI with continuous submergence, using part of the water saved when switching from continuous submergence to 7-day intervals to supply irrigation water to this larger service area.

The data from Tables 3 and 4 indicate that the greatest water use efficiency in rice production is achieved by adopting a schedule with 7-day intervals with SRI crop management. The total need for water according to this irrigation regime is 72% less than the water consumed with the first regime (continuous submergence), and 44% as much water as with the second method (3-day irrigation intervals).

If half of the water that could be saved with SRI and interval irrigation was used in other sectors, this would still leave enough water to add, in principle, 28,100 or 46,650 hectares to the national irrigated area, using 3-day or 7-day irrigation intervals, respectively. These respective expansions would increase the area by 20% or 37%, i.e., to 155,980 or 174,450 hectares. Given the yields achieved in our trials, the respective expansions would raise Iraq's national rice production to 1,095,000 tonnes or 907,000 tonnes, almost double the present production.

With the current cultivation methods and continuous flood irrigation, only about 7.5% of Iraq's current national consumption of milled rice is covered by national production. With SRI and 3-day irrigation intervals, the national production would increase enough to cover 27% of domestic consumption and save about 285 million dollars currently expended on rice imports. Some of this amount of money could be allocated for developing the efficiency of the rice sector, e.g., lining traditional irrigation channels and also drainage channels inside fields to reduce water losses during the crop season.

This scenario is premised on a national objective of achieving the highest land productivity, which is possible with the SRI methods used under the second irrigation regime (3-day irrigation intervals). However, as we have seen, water productivity is the greatest when SRI methods are used with the third regime (7-day intervals). Which goal should be given priority is up to national policy-makers. Our analysis shows the tradeoff to be considered between maximizing land productivity vs. water productivity.

It would definitely take more institutional capacity and greater policy revision to achieve the latter objective; however, with water being such a scarce and severely constraining resource in Iraq, the option of achieving the highest water productivity is more attractive here than it is perhaps in other countries.

## 4. Conclusions

Today, irrigation is the largest single consumer of freshwater on the planet. Competition for water from other sectors will force irrigation systems to operate with more concern for accommodating operations to water scarcity than previously. By reducing the

consumption of irrigation water, deficit irrigation can aid various countries and populations in coping with situations where their water supply is restricted.

SRI based on alternate wetting and drying has emerged as an important agronomic strategy for dealing with water-scarce situations. Proponents of AWD report that this irrigation methodology can reduce the water requirements for irrigated rice production with little or no reduction in the yield, producing the benefit of water saving [14]. With SRI, which is a more complete and integrated strategy for managing plants, soil, water and nutrients, yields can be increased at the same time that water consumption is reduced. This gives farmers more incentive to cooperate in making changes in their age-old cultivation practices.

- Our cost–benefit analysis indicated that the best way to achieve the highest net economic return is with a 3-day-interval irrigation combined with SRI practices. This irrigation protocol produced a net income ha$^{-1}$ of \$2637, compared with \$1950 ha$^{-1}$ with continuous submergence, and \$1702 ha$^{-1}$ with 7-day intervals.
- Further analysis similarly found that the irrigation regime of 3-day intervals with SRI methods had the highest internal rate of return among the three irrigation alternatives evaluated.
- Investing in the cultivation of irrigated rice in Iraq with SRI methods and 3-day irrigation intervals should generate enough net benefit and revenue to compensate the government for the expense involved in transitioning to such an irrigation regime.
- Such investment will not only enhance food security, but water security as well. In addition, there can be some improvement in the quality of soil and water resources when deficit irrigation is practiced with SRI methods, using less chemical fertilizer and relying more on organic materials for maintaining soil fertility.
- The greatest water productivity would be achieved by using SRI methods with 7-day irrigation intervals, producing 236 kg of rice per cubic meter of irrigation water. This compared very favorably with the 178 kg of rice per m$^3$ of water produced with 3-day irrigation intervals (25% less), and the 74 kg of rice with continuous submergence of the paddy fields (69% less). These three comparisons include SRI crop management. The productivity gains would be even greater if compared with current farmer practice.
- With the water saved by moving to the 7-day-interval irrigation, other productive activities could be undertaken and supported that benefit farmers as well as other Iraqis, and the country's rice production itself would become more sustainable, as well as greater. With increasing water scarcity, rice production by current means is becoming less and less profitable and feasible, which puts Iraqi food security in jeopardy.

From the data and analysis presented here, we conclude that the present practice of continuous flooding of rice fields and crop submergence is a waste of irrigation water. Current prevailing rice-growing practices produce no significant benefits, either economic or technical, compared with SRI alternatives.

**Author Contributions:** Descriptive and quantitative analysis methods: K.A.H. and A.J.M.; data curation, M.K.M., K.A.H. and A.J.M.; writing—original draft preparation M.K.M., K.A.H. and A.J.M. All authors have read and agreed to the published version of the manuscript.

**Funding:** This research was funded by Agricultural Research Office, Ministry of Agriculture, Iraq.

**Acknowledgments:** Al-Mishkhab Rice Research Station, Najaf, provided the administrative support and all the material requirements for conducting the trials, such as water meters, seeds, fertilizers, labor force, and harvesting at maturity.

**Conflicts of Interest:** The funder had no role in the design of the study; in the collection, analyses, or interpretation of data; in the writing of the manuscript; or in the decision to publish the results.

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
