# Peer review of "Water Savings, Yield, and Economic Benefits of Using SRI Methods with Deficit Irrigation in Water-Scarce Southern Iraq"

_agronomy, doi:10.3390/agronomy13061481_

Round 1

Reviewer 1 Report

The system of rice intensification (SRI) is a high water-saving cultivation technology with high grain yield and high water use efficiency. SRI would play positive role to improve rice production in Iraq when water resource is competed with other industries. This study assessed the water productivity, yield potential, and economic benefits of SRI under intermittent irrigation in Iraq. The results are important for guiding rice production with high-yielding and high water use efficiency in Iraq. However, current manuscript is poor writing style. In many parts in this manuscript, the social factors are analyzed in stead of the main studying findings. In addition, many flaws exist in this version.

(1) The abstract is too long and many unrelated results are presented here.

(2) Too much research background is introduced from paragraph 1 to paragraph 4 in introduce part. In this part, authors should present more information to introduce the research progress of SRI, intermittent irrigation regimes, and potential interaction effects of SRI×intermittent irrigation regimes on yield and WUE, et al. 

(3) The experiment design is not scientific and feasible. All data can not be analyzed statistically because there is no biological repetition. Therefore, the all results could be not rigorous.

(4)  The conclusions should be focused on the main findings. Apparently, the conclusions part is redundancy in current version.

(5) Only 11 papers were cited in this manuscripts. Thus, the few citation is difficult to reflect the research progress in this field.

Author Response

Dear reviewers

Many thanks for your comments. reply my corrected article according yours notes. Concerning to the references, I would like to say: I add tow more references, this is what introduction allows. 

All the Best 

Reviewer 2 Report

The current manuscript entitled “Water Savings, Yield, and Economic Benefits from Using SRI Methods with Deficit-Irrigation in Water-Scarce Southern Iraq” by Mohammed et al. evaluated intervals of irrigation in conjunction with the use of SRI methods that could achieve the best economic as well as agronomic returns when growing irrigated rice under the water-deficit conditions of southern Iraq. After a careful review, I found that this manuscript lies within the scope of the Agronomy journal and can be accepted after moderate revision. My specific comments are:

1.      In the title, please write the full form of SRI i.e. System of Rice Intensification. Same in the abstract and introduction too.

2.      Please reduce the abstract to less than 250 words and rewrite it with more focus on the following aspects: research problem, the objective of the work, methodology, major numerical results, their relevance to current work, and usefulness of obtained results.

3.      Poor keyword selection, please rewrite them and avoid those that already appeared in the title.

4.      For units either use the -1 form (line 40) or use the per kg form (line 43). Revise the whole manuscript.

5.      Line 46-49: References are a must for these claims.

6.      It is a must to create subheadings under 2. Materials and Methods. E.g., material collection, study design and operation, data analysis, software, etc.

7.      Move Figure 1 to the appropriate subheading of Materials and Methods.

8.      Line 151: remove the underline from “W”.

9.      The results presented are not replicated, I suggest adding STDEV or ST ERROR.

10.   The conclusion section is too long. Major parts of this section can be shifted to the discussion. Please reduce it to 200 words with specific attention to the overall outcome of this study, limitations, and future directions.

11.   Line 335-364: these sections are not drafted as per MDPI guidelines.

12.   Reference formatting is not as per MDPI.

13.   I have a major concern about the very less number of references used to support a wide range of claims made by authors. The authors must add new and latest references (>30) and support all statements.

Author Response

Dear reviewer 

Many thanks for reviewed this article, I corrected it according to your notes. 

All the Best

Khidhir 

Reviewer 3 Report

Important research results for science and agricultural practice. I appreciate that these are field experiments. However, the manuscript needs improvement. Detailed comments are included in the original text (pdf). General notes: - the abstract should be shortened, - add the Latin name of rice to the keywords, - line 46-48 add reference to literature, - include item [10] in the text, - in what years was the experiment conducted? - in which laboratory the soil analysis was performed, - do you have weather conditions from the nearest weather station? - it is good to calculate the results statistically (I know that this is a one-factor experiment and only three levels of the factor) - no references to literature in the "Discussion" chapter, this needs to be changed, - adapt the list of literature to the requirements of the journal, - 11 books of literature is not enough, there is a lot of literature about rice that has been published recently I read the article with interest, but it needs to be improved before being published in the journal Agronomy

Author Response

Dear reviewer

Many thanks for reviewed our article, I corrected its according your notes.

Concerning of weather conditions: Not have data from nearest station, because not need its for this study.

Concerning of the reference in discussion: The reference placed here to underscore importance of using Alternate Wetting and Drying (AWD) to support our findings in this study. 

Please, told other two reviewers (one and two) this is the finest article included all information to be corrected from notes of them.

All the Best

Khidhir 

Round 2

Reviewer 1 Report

There is no additiional conmments in this revision. It should be note that  some issues did not solve raised by me at R1 verison.

Author Response

Thanks for your comments. We have double-checked your first report and revise them.

Reviewer 3 Report

The manuscript titled "Water Savings, Yield, and Economic Benefits from Using SRI Methods with Deficit-Irrigation in Water-Scarce Southern Iraq" has been improved.
Still, the Results and Discussion chapter does not contain references to the literature. There are only 14 items in the bibliography. It's still not enough.
Please see the latest issues of Agronomy journal. The Methodology and Methods show that these are one-year field experiments. Better if they are two or three year experiments.
Publications must be adapted to the requirements of the journal.
See: https://www.mdpi.com/journal/agronomy/instructions

Author Response

Dear reviewer

So thanks for your comments, attached the corrected article according to your comments.

All the Best

Khidhir
